

# Association between levels of trimethylamine N-oxide and cognitive dysfunction: a systematic review and meta-analysis

Zhendong Ren and LuMeng Mo

Department of Anesthesiology, Pengzhou People's Hospital, Pengzhou City, Chengdu, Sichuan Province, China

## ABSTRACT

**Background**. Previous studies have shown that the differential expression of trimethylamine N-oxide (TMAO) is closely related to the prognosis of cognitive dysfunction, but the conclusion is still controversial. Therefore, we conducted a meta-analysis to investigate the difference of TMAO levels between patients with and without cognitive dysfunction.

**Methods**. From the beginning to January 25, 2024, we search for correlational studies on PubMed, Embase, Scopus, the Cochrane Library, and Web of Science. We sought to evaluate the association between TMAO, a major gut microbial metabolite, and cognitive function.

**Results**. To investigate the differences in TMAO values between patients with and without cognitive dysfunction. The database search identified 229 studies. After applying exclusion criteria, seven studies involving 1,675 people (352 patients with cognitive dysfunction and 1,323 controls) were finally included in the meta-analysis. Compared with the control group, patients with cognitive dysfunction had significantly higher TMAO (standard mean differences (SMD): 1.21, 95% confidence interval (CI) [0.53–1.89], $p = 0.0005$, $I^2$ : 96%).

**Conclusion**. Patients with cognitive dysfunction have elevated TMAO levels, and TMAO levels are correlated with cognitive dysfunction.

## INTRODUCTION

Cognitive dysfunction is characterized by reduced or impaired mental and/or intellectual function, affecting cognitive areas like memory, executive function, attention, language, and visuospatial skills. Severity of cognitive dysfunction is categorized into mild cognitive impairment (MCI) and dementia (*Zhou et al., 2023*). Numerous research studies and empirical observations indicate that mental faculties tend to decline as individuals age, making older adults more vulnerable to cognitive issues like cognitive dysfunction (*Salthouse, 1996*; *Van Hooren et al., 2007*). The cognitive dysfunction not only affects the individuals themselves, but also has far-reaching consequences for their families, healthcare

Corresponding author
LuMeng Mo,
miaowa_zhongzi@163.com

providers, and the broader society, leading to substantial physical, psychological, and financial challenges (*Dan, Blazer & Catharyn, 2015*; *Yang et al., 2020*).

The gut microbiome communicates bidirectionally with the brain through neural, gut, immune, and hormonal pathways, a phenomenon known as the gut-brain axis (GBA) (*Grenham et al., 2011*; *Clarke et al., 2013*; *Sherwin et al., 2016*). The term gut-brain axis was first coined in the 1980s (*Sherwin et al., 2016*). Changes in the composition of the gut microbiota can lead to dysbiosis and subsequent functional and metabolic alterations (*Kelly et al., 2016*). Research from clinical and experimental studies has demonstrated dysregulation of the gut microbiota plays an important role in the pathophysiology of a variety of neurodegenerative diseases and cardiovascular diseases (*Kelly et al., 2016*; *Chidambaram et al., 2022*; *Ma et al., 2019*; *Xiao et al., 2020*).

The pathogenesis of cognitive dysfunction is intricate. Recent research on the microbiota-gut-brain axis has revealed a significant connection between metabolites of intestinal flora and cognitive function. Trimethylamine N-oxide, an intestinal metabolite, is produced from choline, L-carnitine, betaine, and other choline-rich compounds found in the diet. These precursors are metabolized by gut microbes and enzymes into TMA, which then travels to the liver *via* the portal circulation and is further transformed into trimethylamine N-oxide (TMAO) by hepatoflavin monooxygenase (*Yang et al., 2019*). Elevated TMAO levels are associated with cardiovascular disease, neurological and sleep disorders, metabolic diseases, and cancer (*Janeiro et al., 2018*; *Yang et al., 2022*; *Lee et al., 2021*; *Lemaitre et al., 2021*; *Praveenraj et al., 2022*).

Increasing evidence suggests that neuroinflammation serves as a crucial mediator of normal cognitive aging (*Raymond, 2010*; *Arthur et al., 2011*). Elevated levels of TMAO may initiate an inflammatory cascade, particularly through the activation of the NF-kB pathway. This activation increases the risk of neuroinflammation when the integrity of the blood–brain barrier (BBB) is compromised, ultimately leading to cognitive decline (*Lanz et al., 2022*; *Benjamin et al., 2020*; *Ohad et al., 2018*). Furthermore, elevated TMAO levels may accelerate brain aging by promoting mitochondrial damage and increasing superoxide production, thereby exacerbating age-related cognitive dysfunction (*Dang et al., 2018*; *Daisuke et al., 2018*; *Daisuke et al., 2020*). Additionally, TMAO may worsen brain aging and cognitive decline in mice by promoting neuronal senescence, disrupting synaptic integrity, and downregulating the expression levels of proteins related to synaptic plasticity and the mechanistic target of rapamycin (mTOR) signaling pathway (*Dang et al., 2018*). Collectively, these findings underscore the detrimental effects of elevated TMAO concentrations on cognitive function. In light of the findings from the aforementioned studies, we conducted the first meta-analysis of published research aimed at assessing the differences in TMAO levels among individuals with and without cognitive dysfunction.

## MATERIALS AND METHODS

### Search strategy

Establish a registration number with PROSPERO prior to the research (registration number: CRD42024504613). A literature search of the Web of Science, Embase, Scopus, PubMed,

and The Cochrane Library was performed from inception to January 25, 2024. "Cognitive Dysfunction" or "Cognitive Dysfunctions" or "Dysfunction, Cognitive" or "Dysfunctions, Cognitive" or "Cognitive Impairments" or "Cognitive Impairment" or "Impairment, Cognitive" or "Impairments, Cognitive" or "Cognitive Disorder" or "Cognitive Disorders" or "Disorder, Cognitive" or "Disorders, Cognitive" or "Mild Cognitive Impairment" or "Cognitive Impairment, Mild" or "Cognitive Impairments, Mild" or "Impairment, Mild Cognitive" or "Impairments, Mild Cognitive" or "Mild Cognitive Impairments" or "Cognitive Decline" or "Cognitive Declines" or "Decline, Cognitive" or "Declines, Cognitive" or "Mental Deterioration" or "Deterioration, Mental" or "Deteriorations, Mental" or "Mental Deteriorations" and "trimethyloxamine" or "trimethylammonium oxide" or "trimethylamine N-oxide" or "TMAO" or "trimethylamine oxide" were the search terms. A comprehensive review of pertinent research citations was carried out. To ensure the accuracy and consistency of the results, two researchers (Zhendong Ren and Lumeng Mo) conducted a database search and summarized the results respectively, and any differences of opinion were discussed to make decisions.

### Inclusion and exclusion criteria

The inclusion criteria were as follows: (1) The exposure group was defined as patients with cognitive dysfunction, and the control group was defined as subjects with normal cognitive function. (2) TMAO values could be extracted in both groups. (3) publish before January 25, 2024; (4) published in English.

Exclusion criteria are as follows: (1) Conference reports, *in vitro* or *in vivo* experiments and reviews; (2) Duplication of publications; (3) Studies with incomplete data or no relevant outcomes.

### Data extraction and quality assessment

Two researchers (Zhendong Ren and Lumeng Mo) independently explored and selected articles based on the specified criteria. They evaluated the articles and extracted data on TMAO levels in patients with cognitive dysfunction and controls, the sample type, the publication year, the study region, the definition of cognitive function, and the TMAO measurement method. In case of discrepancies during data extraction, the authors discussed and reached a consensus. The Newcastle-Ottawa Scale (NOS) was employed to assess the quality of the literature (*Ma et al., 2020*). Each study was evaluated using a nine-item scale, with one point awarded for each completed item. Total scores varied from 0 to 9, with a NOS score of 7 or higher indicating high-quality study results. Detailed scores are presented in Table 1.

### Statistical analysis

All data were obtained from the included studies. For publications stating interquartile ranges and medians, means and standard deviations were estimated using the formula described in the Cochrane Handbook for interquartile ranges. For publications stating median, minimum, and maximum values, means and standard deviations were estimated using the method described by *Stela Pudar, Benjamin & Iztok (2005)*.

Ren and Mo (2025), *PeerJ*, DOI 10.7717/peerj.20000

**Table 1  Newcastle-Ottawa Scale (NOS) adapted for quality rating for the seven included studies.**

| Study | Case definition | Representativeness | Selection of controls | Definition of controls | Comparability | Ascertainment of exposure | Same method | Non response rate | Total |
|---|---|---|---|---|---|---|---|---|---|
| Wei He | 1 | 1 | 0 | 0 | 2 | 1 | 1 | 0 | 6 |
| Li Gong | 1 | 1 | 0 | 0 | 2 | 1 | 1 | 0 | 6 |
| Nongzhang Xu | 1 | 1 | 0 | 0 | 2 | 1 | 1 | 0 | 6 |
| Lufeng Wang | 1 | 1 | 0 | 0 | 2 | 1 | 1 | 0 | 6 |
| Nicholas M Vogt | 1 | 0 | 1 | 1 | 2 | 1 | 1 | 0 | 7 |
| Ali Yilmaz | 1 | 0 | 1 | 1 | 2 | 1 | 1 | 0 | 7 |
| Szu-Ju Chen MD | 1 | 0 | 0 | 1 | 2 | 1 | 1 | 0 | 6 |

**Notes.**

*He et al., 2020*; *Gong et al., 2021*; *Xu et al., 2022*; *Wang et al., 2023*; *Vogt et al., 2018*; *Yilmaz et al., 2020*; *Chen et al., 2020*.

Review Manager 5.4.1 and Stata17 were utilized to estimate all statistical data. Continuous variables were assessed using standard mean differences (SMD), with effect sizes displayed as 95% confidence intervals (CI). Statistical heterogeneity was quantified as $I^2$, where a value of 0% indicates no observed heterogeneity, and higher values indicate increased heterogeneity levels. Specifically, 25% represented low, 50% moderate, and 75% high heterogeneity. In cases of moderate heterogeneity (PQ > 0.1, $I^2 < 50\%$), a fixed effects model was employed; otherwise, a random effects model was used. Data were visually presented through Forest plots. Sensitivity analysis was conducted to identify sources of heterogeneity and ensure result stability. Egger's test and Begg's funnel plot were performed to assess potential publication bias. Statistical significance was defined as $p < 0.05$.

## RESULTS

### Characteristics of the study

A total of 229 studies were initially obtained based on the search strategy (Web of Science = 66, Embase = 86, Scopus = 25, PubMed = 46, Cochrane = 5, other sources=1). After eliminating 97 duplicate studies, 113 articles were excluded based on title and abstract screening as irrelevant. The full texts of the remaining 19 studies were thoroughly reviewed, leading to the exclusion of 12 studies due to insufficient data. Ultimately, seven studies (*Vogt et al., 2018*; *Chen et al., 2020*; *He et al., 2020*; *Yilmaz et al., 2020*; *Gong et al., 2021*; *Xu et al., 2022*; *Wang et al., 2023*) met the criteria and were included in the meta-analysis. The study screening procedure is shown in Fig. 1. The seven studies included a total of 1,675 patients, with 352 in the cognitive dysfunction group and 1,323 in the control group. All studies evaluated TMAO values in both groups, with five studies conducted in China and two in the United States. Sample types included fasting plasma in five studies, cerebrospinal fluid in one study, and urine in one study. The studies were published between 2018 and 2023. Cognitive function was defined using MoCA in three studies, MMSE in two studies, NIA-AA diagnostic criteria in one study, and NINCDS and ADRDA diagnostic criteria in one study. The characteristics of the articles included were summarized in Table 2.

### Meta-analysis of cognitive function and TMAO levels

Based on the findings of our meta-analysis, individuals with cognitive dysfunction exhibited notably elevated TMAO levels in comparison to control subjects (SMD: 1.21, 95% CI [0.53–1.89], $p = 0.0005$). The studies included displayed substantial heterogeneity ($I^2 = 96\%$, $P < 0.00001$), leading to the utilization of a random-effects model for assessing the treatment effect as shown in Fig. 2.

### Subgroup analysis

Subgroup analyses were conducted to address the high heterogeneity among the included literature. The analysis specifically examined study country and tissue sample type. Out of the studies included, two were carried out in the United States, while the rest were conducted in China. In the subgroup analysis by study country, (China (SMD: 1.39; 95% CI [0.39–2.39]; $I^2 = 97\%$), United States (SMD: 0.75; 95% CI [0.52–0.98]; $I^2 = 0\%$)) as shown in Fig. 3. Regarding tissue sample type, one study used cerebrospinal fluid, one

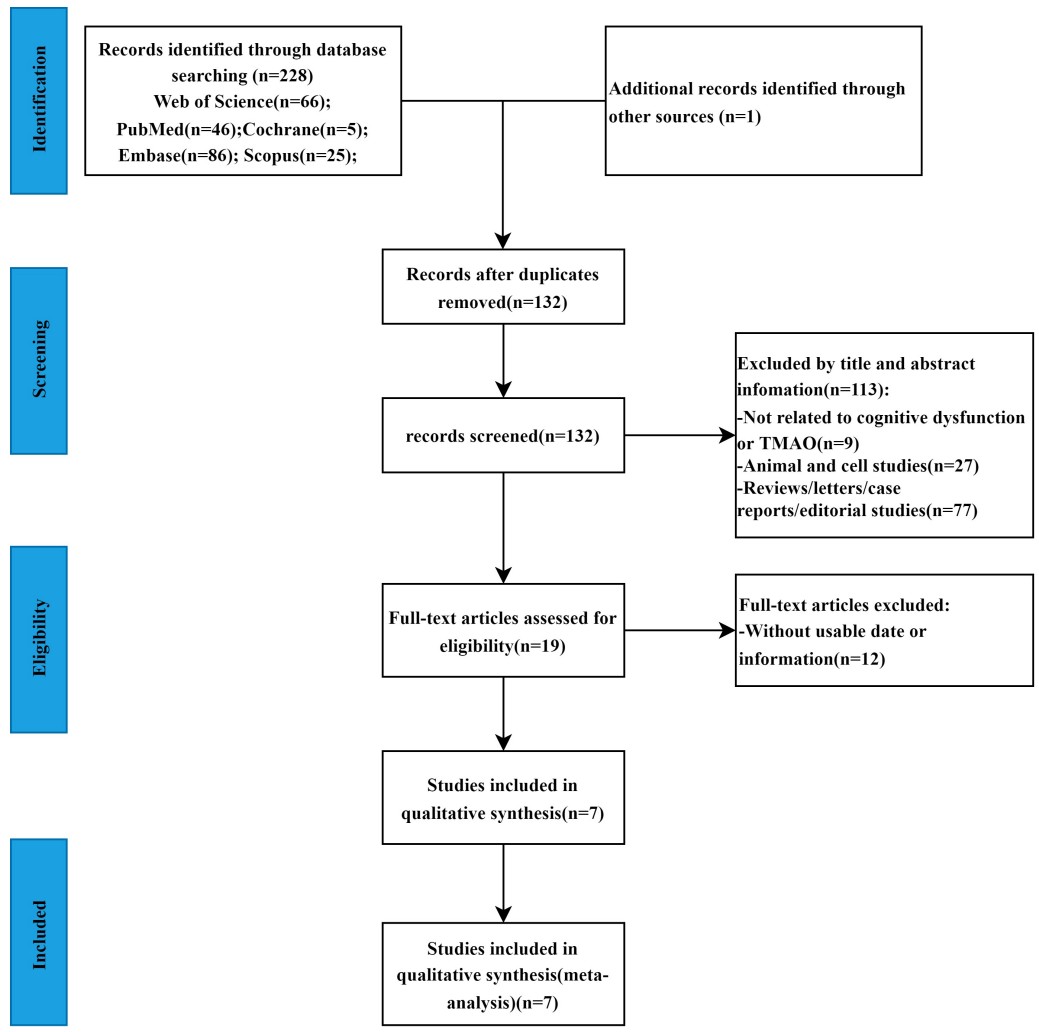

**Figure 1** **The flowchart of the meta-analysis.** Characteristics of the studies included in the meta-analysis.

study used urine, and the others used plasma. In the subgroup analysis of tissue sample type, (plasma (SMD: 1.39; 95% Cl [0.39–2.39]; $I^2 = 97\%$), other (SMD: 0.75; 95% Cl [0.52–0.98]; $I^2 = 0\%$)), as shown in Fig. 4.

## Publication bias and sensitivity analysis

This meta-analysis conducted a thorough assessment of publication bias by utilizing Begg's funnel plot and Egger's test. The results of Begg's funnel plot ($P = 0.133$) and Egger's test ($P = 0.341$) indicated that there was no significant publication bias influencing the outcomes of the meta-analysis. as shown in Fig. 5. Additionally, Fig. 6 depicts the sensitivity analysis carried out, demonstrating the robustness of the meta-analysis results even when each study was removed individually.

## DISCUSSION

In this systematic review and meta-analysis, involving seven studies with 1,675 individuals, we aimed to investigate the relationship between intestinal flora metabolites and cognitive dysfunction by examining the association of the major intestinal flora metabolite TMAO with cognitive dysfunction. Our findings revealed that TMAO levels were notably elevated in patients with cognitive dysfunction compared to control groups. Despite there was considerable heterogeneity among the studies, sensitivity and subgroup analyses were conducted in this meta-analysis to investigate the sources of heterogeneity. Our results indicate that the findings are stable. This heterogeneity may be attributed to variations in measurement techniques or to different molecular mechanisms involved in the disease. In the context of the neurodegenerative disease Alzheimer's disease (AD), TMAO promotes the aggregation of abnormal proteins by disrupting their normal folding and clearance mechanisms (*Allan, 2021*). An increase in TMAO levels leads to the accumulation of amyloid-beta (A$\beta$) peptides and hyperphosphorylated tau protein (p-tau) in neuronal cells, representing a major pathogenic mechanism in AD patients (*William, 2010*). In elderly patients with cardiovascular disease (CVD), elevated TMAO levels are independently associated with the frailty index, which involves mitochondrial dysfunction and chronic inflammatory pathways (*Wei et al., 2020*). Currently, there is no unified international standard for TMAO detection, and the varying measurement methods employed in different studies may contribute to the observed heterogeneity in this meta-analysis.

TMAO is implicated in various biological dysfunctions, including oxidative stress, disruption of the blood–brain barrier, diminished synaptic plasticity, inflammation, mitochondrial dysfunction, and abnormal protein aggregation. These dysfunctions are pivotal to the pathogenesis of cognitive dysfunction (*Haihua et al., 2025*). Research indicates that TMAO accelerates brain aging and cognitive decline by promoting neuronal senescence while exacerbating neuroinflammation and oxidative stress (*Manoj et al., 2020*). The excessive production of reactive oxygen species (ROS) results in DNA damage, lipid peroxidation, and protein oxidation, collectively triggering apoptosis and cognitive dysfunction (*Giovanna et al., 2008*). Furthermore, TMAO disrupts the integrity of the BBB and impairs synaptic plasticity. TMAO has been shown to compromise the structural integrity and functionality of the BBB, thereby diminishing synaptic plasticity and resulting in cognitive deficits. Concurrently, TMAO disrupts the BBB, facilitates the accumulation of neurotoxic molecules within the brain, and induces oxidative stress and neuroinflammation (*Lesley et al., 2021*). The activation of inflammatory signaling pathways further contributes to neuronal damage and cognitive decline (*Ming-Liang et al., 2017*). The pathogenic effect of TMAO appears to be concentration-dependent, with studies indicating that physiologically relevant concentrations of TMAO positively influence BBB integrity and cognition *in vivo* (*Lesley et al., 2021*). This beneficial effect of TMAO starkly contrasts with earlier research that highlighted the detrimental effects of TMAO exposure at high concentrations or under non-physiological conditions. The gut microbiota's communication through the gut-brain axis is intricate, and this bidirectional regulation of TMAO may have led to the oversight of numerous negative studies due to challenges in

Ren and Mo (2025), *PeerJ*, DOI 10.7717/peerj.20000

**Table 2  Characteristics of the studies included in the meta-analysis.**

| Study | Sample type | Year | Region | Cognitive dysfunction definition | TMAO measurement | Cognitive dysfunction | | Control | | NOS |
|---|---|---|---|---|---|---|---|---|---|---|
| | | | | | | N | Age (years) | Age (years) | N | |
| Wei He | plasma | 2020 | China | MMSE | UPLC-MS | 33 | NA | NA | 418 | 6 |
| Li Gong | plasma | 2021 | China | MoCA | NA | 66 | NA | NA | 66 | 6 |
| Nongzhang Xu | plasma | 2022 | China | MoCA | HPLC-MS | 74 | $63.58 \pm 8.5$ | $60.10 \pm 8.22$ | 179 | 6 |
| Lufeng Wang | plasma | 2023 | China | MoCA | LC-MS | 67 | $77.40 \pm 7.88$ | $72.21 \pm 9.76$ | 243 | 6 |
| Nicholas M Vogt | CSF | 2018 | USA | NIA-AA diagnostic criteria | UHPLC-MS | 75 | $72.5 \pm 8.5$ | $61.9 \pm 7.9$ | 335 | 7 |
| Ali Yilmaz | Urine | 2020 | USA | NINCDS and ADRDA diagnostic criteria | LC-MS | 30 | $78.80 \pm 9.18$ | $79.12 \pm 6.28$ | 29 | 7 |
| Szu-Ju Chen MD | plasma | 2020 | USA | MMSE | LC-MS | 14 | NA | NA | 46 | 6 |

**Notes.**

CSF, cerebrospinal fluid; HPLC-MS/MS, high-performance liquid chromatography-tandem mass spectrometry; UPLC-MS/MS, ultra-performance liquid chromatography with online tandem mass spectrometry; LC-MS, Liquid chromatography–mass spectrometry; UHPLC-MS, ultrahigh performance liquid chromatography tandem mass spectrometry; LC-MS/MS, liquid chromatography tandem mass spectrometry; NINCDS, National Institute of Neurological and Communicative Disorders and the Stroke; ADRDA, Alzheimer's Disease and Related Disorders Association; NIA-AA, National Institute on Aging–Alzheimer's Association workgroup; MoCA, The Montreal Cognitive Assessment; MMSE, Mini-Mental State Examination.

*He et al., 2020*; *Gong et al., 2021*; *Xu et al., 2022*; *Wang et al., 2023*; *Vogt et al., 2018*; *Yilmaz et al., 2020*; *Chen et al., 2020*.

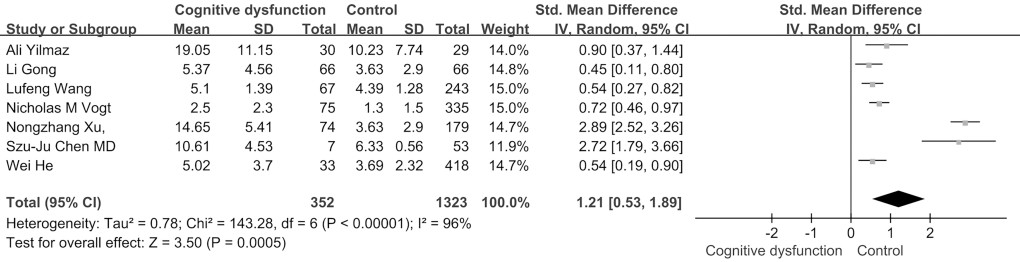

**Figure 2** **Forest plots for cognitive function and TMAO levels.** Forest plot displaying differential levels of TMAO between individuals with and without cognitive dysfunction. Notes: *Yilmaz et al., 2020*; *Gong et al., 2021*; *Wang et al., 2023*; *Vogt et al., 2018*; *Xu et al., 2022*; *Chen et al., 2020*; *He et al., 2020*.

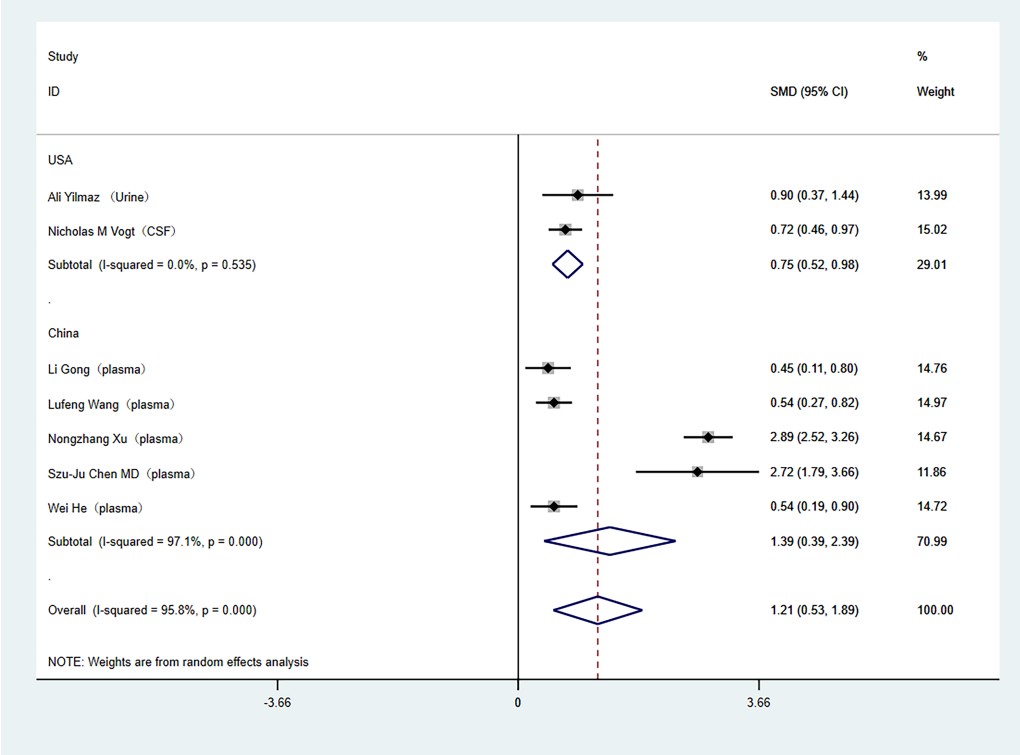

**Figure 3** **Subgroup analyses.** Subgroup analysis of the comparison between TMAO levels according to different region groups. Notes: *Yilmaz et al., 2020*; *Vogt et al., 2018*; *Gong et al., 2021*; *Wang et al., 2023*; *Xu et al., 2022*; *Chen et al., 2020*; *He et al., 2020*.

publication. Although Begg's test and Egger's test were employed to evaluate publication bias in this meta-analysis, the small number of included studies resulted in no significant difference between the two groups. Consequently, the sensitivity of the test is significantly diminished, potentially obscuring the actual presence of bias.

Our study demonstrated that patients with cognitive dysfunction exhibit elevated levels of TMAO compared to individuals without cognitive dysfunction. Furthermore, reducing

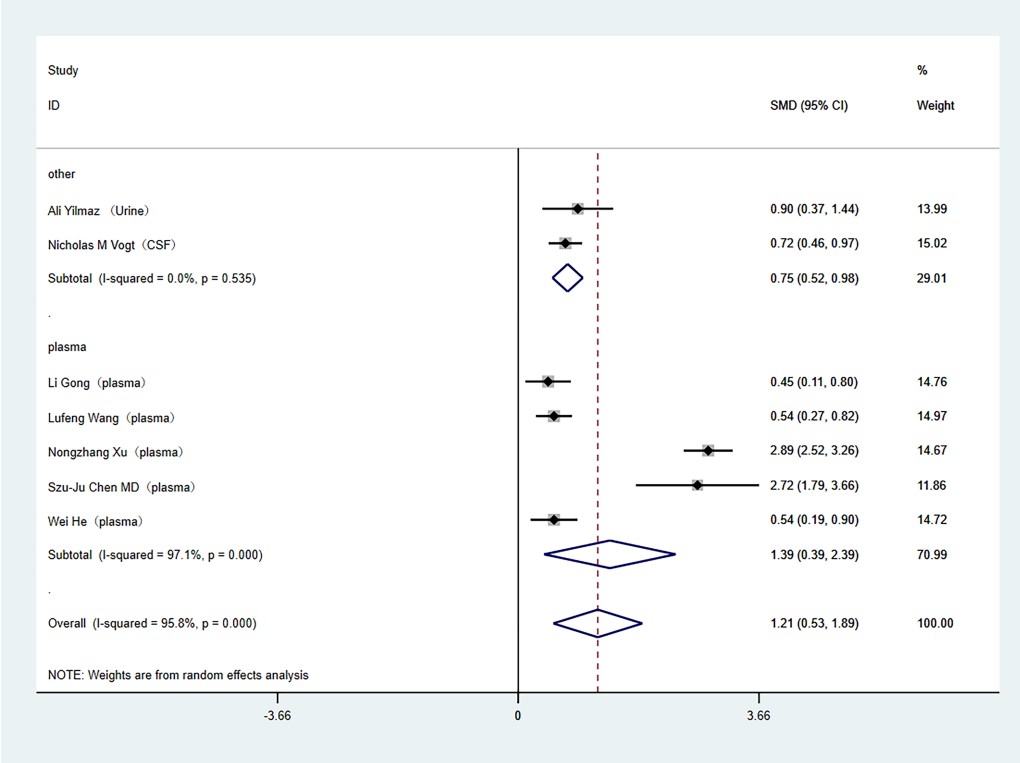

**Figure 4 Subgroup analyses.** Subgroup analysis of the comparison between TMAO levels according to different sample type groups. Notes: *Yilmaz et al., 2020*; *Vogt et al., 2018*; *Gong et al., 2021*; *Wang et al., 2023*; *Xu et al., 2022*; *Chen et al., 2020*; *He et al., 2020*.

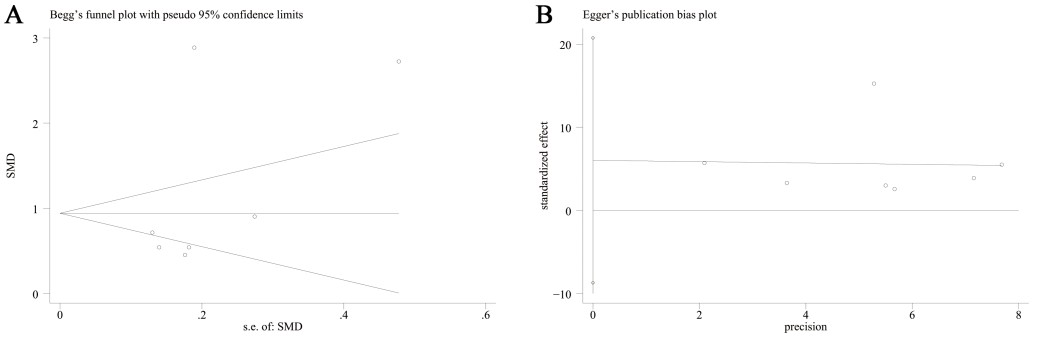

**Figure 5 Begg's funnel plot and Egger' publication bias plot.** (A) Begg's funnel plot of the differential levels of TMAO between individuals with and without cognitive dysfunction; (B) Egger' publication bias plot of the differential levels of TMAO between individuals with and without cognitive dysfunction.

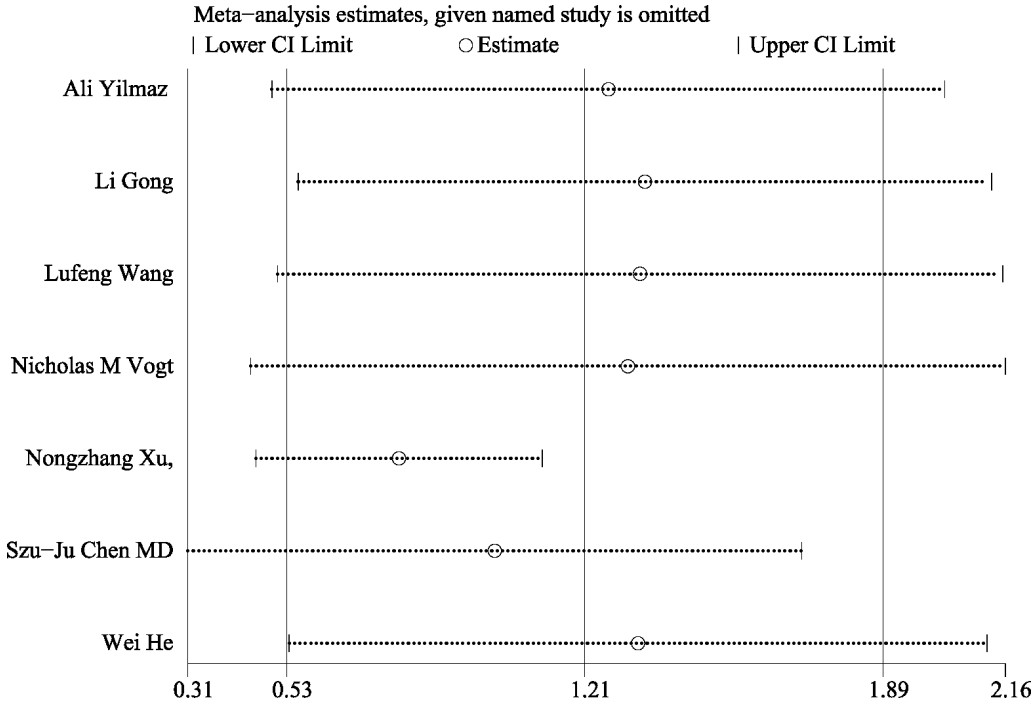

**Figure 6  Sensitivity analysis.** Sensitivity analysis for the differential levels of TMAO between individuals with and without cognitive dysfunction. Notes: *Yilmaz et al., 2020*; *Gong et al., 2021*; *Wang et al., 2023*; *Vogt et al., 2018*; *Xu et al., 2022*; *Chen et al., 2020*; *He et al., 2020*.

TMAO levels may represent a promising intervention strategy for improving cognitive function. Effective interventions aimed at decreasing TMAO production or enhancing its clearance could potentially prevent the onset of cognitive dysfunction. This study serves as a valuable reference for the clinical evaluation of the impact of gut microbial metabolites on cognitive function. The meta-analysis presented in this study is subject to several limitations. Firstly, the sample size was relatively small, and the included projects were limited to studies conducted in China and the United States. Secondly, there was a notable level of heterogeneity in outcome measures, which could be influenced by various factors. Lastly, the potential for bias exists as only English literature was included in the literature search.

## CONCLUSION

In this systematic review and meta-analysis, we found that TMAO levels are associated with cognitive dysfunction. Further research is necessary to explore the role and function of TMAO within the gut-brain axis.

### Funding
The authors received no funding for this work.

### Competing Interests
The authors declare there are no competing interests.

### Author Contributions
- Zhendong Ren conceived and designed the experiments, performed the experiments, analyzed the data, prepared figures and/or tables, and approved the final draft.
- LuMeng Mo conceived and designed the experiments, performed the experiments, analyzed the data, authored or reviewed drafts of the article, and approved the final draft.

### Data Availability
This is a systematic review/meta-analysis.

### Supplemental Information
Supplemental information for this article can be found online at http://dx.doi.org/10.7717/peerj.20000#supplemental-information.

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
