# Peer review of "Association between levels of trimethylamine N-oxide and cognitive dysfunction: a systematic review and meta-analysis"

_PeerJ, doi:10.7717/peerj.20000_

## Round 0.1 · original submission · Major Revisions

**Language Note:** The review process has identified that the English language must be improved. PeerJ can provide language editing services - please contact us at [email protected] for pricing (be sure to provide your manuscript number and title). Alternatively, you should make your own arrangements to improve the language quality and provide details in your response letter. – PeerJ Staff

·

Basic reporting

This systematic review summarizes the results of clinical studies investigating the effects of trimethylamine N-oxide on cognitive function.

I would like to offer several suggestions for improving this manuscript.

1. Include participant ages in Table 1, as age is a key variable in cognitive decline and essential for interpreting the sample characteristics.

2. Provide a table detailing the biases considered in the meta-analysis to ensure methodological transparency and allow for a more robust evaluation of the study’s validity.

Experimental design

-

Validity of the findings

-

Reviewer 2 ·

Basic reporting

This article uses a systematic review and meta-analysis to explore the association between Trimethylamine N-oxide (TMAO) levels and cognitive dysfunction.

While the study ambitiously aims to examine the relationship between TMAO levels and cognitive impairment in depth, its execution falls short in several critical areas. Most notably, the authors fail to adequately elucidate the molecular mechanisms underlying the relationship between TMAO and cognitive impairment. This omission undermines the scientific depth and credibility of the work significantly.

In summary, while the article addresses a significant topic – the potential link between Trimethylamine N-Oxide (TMAO) levels and cognitive dysfunction – it falls short of delivering substantial contributions to the field. The paucity of detailed exploration into fundamental molecular mechanisms, coupled with an analytical approach that is merely superficial in nature, results in a significant weakening of the study. Furthermore, the absence of innovative insights or a novel analytical perspective serves to further diminish its relevance. Consequently, the work under review fails to provide meaningful advancements or enrich the scientific discourse on this critical subject.

Experimental design

The study's lack of focus on biochemical pathways, such as the role of TMAO in inflammation, oxidative stress, or vascular dysfunction, key mediators that could potentially link TMAO to cognitive decline, further limits its contribution to the field. By glossing over these essential elements, the authors miss an opportunity to establish a comprehensive understanding of the topic. Consequently, while the subject matter is significant, shortcomings in analysis and mechanistic explanation render the work underwhelming overall.

It is imperative that studies of this nature contribute a novel perspective or analysis to the field, rather than merely compiling and reiterating existing findings. Systematic reviews and meta-analyses have the capacity to yield significant insights by identifying patterns, gaps, or inconsistencies in the extant literature; however, this endeavour necessitates a degree of depth and analytical rigour that challenges authors to engage critically with the data.

Validity of the findings

This article is lacking in original analytical content. The authors have primarily elected to provide summaries of results that have been previously published, and they have not provided fresh interpretations of the data or made any significant contributions to the existing discussion.

Effective research demands more than mere replication; it requires authors to synthesise findings in a manner that elucidates new connections, refines hypotheses, or proposes actionable directions for future studies.

This fundamental intellectual endeavour appears to be absent in this instance, thereby diminishing the value of the paper and its potential impact on the field.

·

Basic reporting

Language and Clarity:
• Phrasing such as “to be published in English” (line 68) should be revised to “published in English.”

Literature and Background:
• The introduction provides sufficient background on cognitive dysfunction and TMAO, with appropriate references supporting the rationale for the study.
• The literature review adequately highlights the knowledge gap regarding the relationship between TMAO and cognitive dysfunction.

Structure and Formatting:
• The manuscript adheres to standard scientific structure: Abstract, Introduction, Methods, Results, Discussion, Conclusion.
• Figures and tables are well-integrated and appropriately referenced in the text.
• Figure captions should be more detailed, and the individual panels within each figure need clearer labeling and explanation. The figures should be self-explanatory, as their current presentation may be difficult for readers to interpret

Data Availability:
• Raw data presentation appears compliant.

Experimental design

Scope and Relevance:
• The research question is relevant and meaningful, targeting an emerging area within neurogastroenterology.
• The rationale for conducting a meta-analysis is well-justified.

Methodology:
• Search strategy is detailed and aligned with systematic review standards. Registration on PROSPERO (line 51) is appropriately noted.
• The inclusion/exclusion criteria are valid, though the description would benefit from clearer formatting (lines 66–70).
• The Newcastle-Ottawa Scale (NOS) is correctly applied for risk of bias assessment, but the corresponding scores in Table 1 need brief interpretation in the main text.

Reproducibility:
• Methods for statistical analysis are adequately described, including heterogeneity testing, subgroup analysis, and bias assessment.
• The transformation of data into standardized mean differences is noted, but more clarity is needed on how mean ± SD was derived from studies using medians or other central tendencies (lines 76–77).

Validity of the findings

Statistical Soundness:
• The meta-analysis yields a statistically significant association between elevated TMAO levels and cognitive dysfunction (SMD: 1.21, 95% CI 0.53–1.89).
• Heterogeneity is high (I² = 96%), which weakens the strength of pooled conclusions. Subgroup analysis helps, but does not sufficiently resolve heterogeneity.

Data Transparency:
• Begg’s funnel plot and Egger’s test suggest no publication bias.
• Sensitivity analysis is thorough and supports the robustness of the findings.

Conclusions:
• While the results support the hypothesis, the conclusion overreaches in describing TMAO as a biomarker or causal agent without experimental or longitudinal support.
• The study provides a correlation, not a causative link. This should be emphasized in both the discussion and conclusion.
• Suggest the authors temper their conclusions and avoid terms like “important role” unless supported by mechanistic studies.

Additional comments

Strengths:
• Timely topic addressing the gut-brain axis.
• Comprehensive literature search and clear inclusion/exclusion criteria.
• Adequate handling of bias and heterogeneity through appropriate statistical tools.

Weaknesses:
1. High heterogeneity across studies limits the generalizability of findings. Authors should explore additional sources of heterogeneity (e.g., age, disease type, study design).
2. Conclusion extends beyond the data—TMAO is described as a harmful metabolite with mechanistic effects on cognition, which goes beyond what correlational meta-analysis can show.
3. Insufficient detail on the transformation of data and assumptions made in computing pooled effect sizes.

Reviewer 4 ·

Basic reporting

This article presents a meta-analysis exploring the relationship between TMAO levels and cognitive dysfunction, assessed in various types of human samples, including plasma, urine, and cerebrospinal fluid (CSF). Across all included studies, TMAO levels were consistently elevated in individuals with mild cognitive impairment (MCI) compared to the respective control groups. While the study addresses a timely and relevant topic, several aspects require clarification or improvement to strengthen the manuscript.

1. The introduction focuses mainly on two references (articles 12 and 13) to support the harmful effects of TMAO on cognitive function. However, there is a broader body of literature describing TMAO’s impact on cognition and its mechanisms of action in the central nervous system. The authors are encouraged to expand this section to include additional relevant studies.

2. Additionally, the manuscript briefly mentions controversy regarding the role of TMAO in cognition. Are there studies showing neutral or even beneficial effects? If so, these should be included and discussed in the discussion to provide a balanced background and better justify the rationale for this meta-analysis.

3. TMAO has been implicated in neurodegenerative disorders, including Alzheimer’s disease. This connection should be mentioned either in the introduction or the discussion to contextualize the relevance of TMAO beyond MCI.

4. Much of the content in the discussion (particularly regarding the microbiota–gut–brain axis and the microbial origin of TMAO) would be more appropriate in the introduction. Consider restructuring to improve flow and scientific logic.

Experimental design

5. It would be beneficial to comment on the comparability of TMAO quantification methods across the studies. Are the methodologies (e.g., UPLC-MS, HPLC-MS, LC-MS, etc.) standardized? Can you reference literature that has addressed the inter-study comparability of TMAO or similar metabolites to support the methodology you have employed?

Validity of the findings

6. The authors state that TMAO levels vary across different ages. It would be helpful to indicate the average age of participants in each of the included studies. Were the control and cognitive impairment groups age-matched in these studies? If not, how was age accounted for in the meta-analysis? Please clarify whether age was used as a covariate or corrected for statistically.

7. While the trend across studies suggests that individuals with MCI consistently show higher TMAO levels than controls, regardless of the region or biological sample (plasma, CSF, or urine), it is questionable to group all these measurements together in a single analysis. The biological significance of TMAO levels may differ between sample types, and combining them could mask relevant distinctions. The rationale for aggregating data across these biologically diverse sources should be clearly explained.

8. The authors should clarify the reasons for excluding studies with inconclusive or negative results. If such studies were excluded simply because they did not find significant differences in TMAO levels between control and cognitively impaired individuals, this introduces bias and undermines the objectivity of the meta-analysis. Please provide details on the exclusion criteria and, if applicable, discuss the potential impact of publication bias.

Additional comments

Figures and Presentation
• Figure 2: The column headers ("Control" and "Cognitive Dysfunction") should be aligned with the data rows ("Mean," "SD," and "Total") for clarity.
• Figures 3 and 4: It would be helpful to indicate the sample type (plasma, CSF, or urine) used in each study directly within the figure or its legend.
• Figure 6: The lines are very thin, and the labels are difficult to read due to their small size. Please enhance the visibility and layout of this figure.
• Figure Legends: All figures currently lack descriptive legends. Each figure should have a clear legend explaining the data presented.

Minor Comments
1. Abstract (Line 15): The abstract states that the authors "explore the correlation between TMAO and cognitive function." However, the analysis actually compares TMAO levels in individuals with and without cognitive impairment, without directly assessing cognitive performance levels. Please rephrase for accuracy.

2. Line 75: The sentence "In the case of differences in data extraction, discuss together to make decision" is grammatically incorrect. Consider revising it to: "In case of discrepancies during data extraction, the authors discussed and reached a consensus."

3. Conclusions (Line 162): The authors refer to TMAO as a “harmful microbiota metabolite.” This statement is too strong given the existing scientific debate. A more appropriate phrasing would be: “These findings suggest that TMAO may act as a harmful microbiota metabolite associated with cognitive dysfunction.”

---

## Round 0.2 · accepted · Accept

Two reviewers agree, and I concur, that the revised version adequately addresses all the reviewer comments, and that the manuscript is greatly strengthened as a result of these revisions. The manuscript is now ready for publication.

·

Basic reporting

I have no comments or suggestions for improving this manuscript.

Experimental design

No comments

Validity of the findings

No comments

Additional comments

No comments

Reviewer 4 ·

Basic reporting

The authors have addressed all the concerns raised within the point-by-point revision. They have expanded and strengthened the introduction and discussion, clarified methodological and analytical choices, added missing information (e.g., participant ages, exclusion criteria), and revised the figures and legends for clarity. Based on the constructive revisions provided, I am satisfied with the authors' responses and consider the manuscript improved accordingly.

Experimental design

The authors have clarified the questions regarding the experimental design.

Validity of the findings

The authors have clarified the questions regarding the validity of the findings.

Additional comments

I do not have any additional comments.